**DOI: 10.1038/ncomms12349**　　**OPEN**

# Climate change velocity underestimates climate change exposure in mountainous regions

Solomon Z. Dobrowski[1] & Sean A. Parks[2]

Climate change velocity is a vector depiction of the rate of climate displacement used for assessing climate change impacts. Interpreting velocity requires an assumption that climate trajectory length is proportional to climate change exposure; longer paths suggest greater exposure. However, distance is an imperfect measure of exposure because it does not quantify the extent to which trajectories traverse areas of dissimilar climate. Here we calculate velocity and minimum cumulative exposure (MCE) in degrees Celsius along climate trajectories for North America. We find that velocity is weakly related to MCE; each metric identifies contrasting areas of vulnerability to climate change. Notably, velocity underestimates exposure in mountainous regions where climate trajectories traverse dissimilar climates, resulting in high MCE. In contrast, in flat regions velocity is high where MCE is low, as these areas have negligible climatic resistance to movement. Our results suggest that mountainous regions are more climatically isolated than previously reported.

[1] Department of Forest Management, College of Forestry and Conservation, University of Montana. Missoula, Montana 59812, USA. [2] Aldo Leopold Wilderness Research Institute, Rocky Mountain Research Station, US Forest Service. Missoula, Montana 59812, USA. Correspondence and requests for materials should be addressed to S.Z.D. (email: solomon.dobrowski@umontana.edu) or to S.A.P. (email: sean_parks@fs.fed.us).

Climate change impacts on biota depend in part on climate change exposure, the degree to which a system is exposed to climate variations over time or space[1,2]. One of the most widely used metrics for estimating exposure is climate change velocity, the direction and rate at which organisms must move to maintain a given climate[3]. Multiple formulations have been proposed for calculating climate velocity, including approaches based on local climate gradients[3] (herein gradient based) and those based on distance to analogue climates (distance based)[4]. These metrics have been used to describe risk of species extinctions[5], climate change responses of marine taxa[6], regional patterns in species endemism[7], Quaternary range shifts of biota[8], the distribution of climate change refugia[9] and climate change exposure under observed and projected future conditions[3,4,10–12]. Velocity has also been used to derive climate trajectories, paths that describe the movement of a climate isopleth over a given period of time[13]. Although velocity estimates are dependent on a number of methodological choices[4,10,11], the relative rankings of velocity have been shown to be quite robust to differences in methodology[4] and show a general pattern of low values (implying longer climate residence times) in regions with high spatial climate heterogeneity (for example, mountainous regions), and high velocity (shorter residence times) in areas of low topographic relief[3,10,12]. Consequently, authors have suggested that flat areas are particularly exposed to climate change while areas of complex terrain may act to buffer climate change impacts by allowing organisms to ameliorate climate shifts via short distance dispersal[10,14–16].

Implicit in the use of velocity is the assumption that climate trajectory length is proportional to exposure; longer paths suggest greater exposure[3,4,10,12,17]. However, distance is an imperfect measure of exposure in the same way it is an insufficient measure of connectivity[18,19]. Climatic connectivity—the ability of a landscape to promote or hinder species movement in response to a changing climate—is also contingent on the costs of moving through areas of dissimilar climate. These costs may preclude migration in areas with strong climate gradients. For instance, the closest climate analogue for a mountain-top may be found on an adjacent mountain-top. The distance may be short between each

point, but the actual exposure to climate differences along the trajectory through the ensuing valley can be large (Fig. 1). In addition, exposure will depend on assumptions about dispersal preferences of organisms. Current distance-based approaches[4,11] to estimating velocity assume that organisms will minimize the distance they travel (Euclidean distance—ED) as opposed to minimizing their exposure to dissimilar climate. If organisms minimize exposure and thus deviate from straight line paths, then velocity based on ED (velocity$_{ED}$) will underestimate rates needed to keep pace with changing climate (Fig. 1).

Here we propose a set of modifications for distance-based velocity based on the assumption that organisms will follow paths that minimize their exposure to dissimilar climates. The distance along this trajectory from source to destination pixel (minimum exposure distance; MED) will be greater than or equal to distances calculated using ED. To account for this assumption, we employ least-cost modelling techniques (see Methods section) applied to isotherms within North America (sensu[20]) for the interval of 1995–2085. We contrast velocity$_{ED}$ with velocity estimates based on MED (velocity$_{MED}$). In addition, we quantify the minimum cumulative exposure (MCE) in °C along each climate trajectory which to our knowledge, is a previously unreported facet of exposure. We find that velocity is weakly related to MCE; both velocity$_{ED}$ and velocity$_{MED}$ underestimate exposure in mountainous regions where even short climate trajectories traverse landscapes with dissimilar climate resulting in high values of MCE. In contrast, velocity is high over flat regions of the continent where MCE is low, as these areas have negligible climatic resistance to movement.

## Results

**Velocity and Minimum Cumulative Exposure.** For North America, the average (geometric mean) climate velocity based on MED (velocity$_{MED}$) was 3.25 km per year (5–95th percentile; 0.28–30.9 km per year). In comparison, average velocity based on ED (velocity$_{ED}$) was 2.68 km per year (0.24–11.8 km per year), which indicates that velocity$_{MED}$ is 21% higher on average than velocity$_{ED}$. The location of the 'nearest' climate analogue varies depending on whether we assume organisms minimize exposure

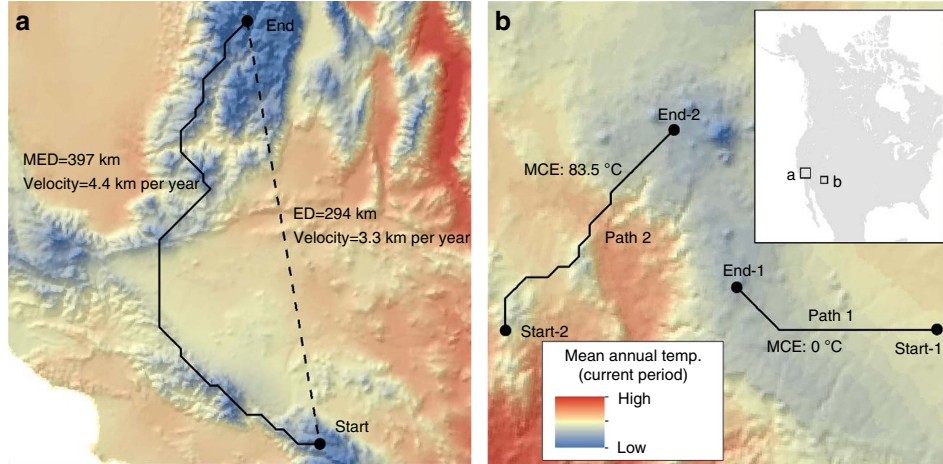

**Figure 1 | Climate trajectories and minimum cumulative exposure.** Climate trajectories are defined by a source pixel (start) with a given temperature under current conditions (1981–2010) and a destination pixel (end) with a similar temperature under future conditions (2071–2100). An example of the difference between climate trajectories defined by Euclidean distance (ED) and minimum exposure distance (MED) for a site within North America (inset) is presented in **a**. MED (solid line) minimizes exposure to climates that are dissimilar to that of the source pixel whereas ED (dashed line) minimizes distance travelled. An example of minimum cumulative exposure (MCE) for two trajectories is presented in **b**. Although the trajectories are similar in length, MCE = 0 for path 1 because the trajectory tracks changes in climate over the study period without traversing temperatures that are dissimilar to that of the source pixel. Path 2 has a high MCE because the trajectory traverses two warm valleys (shading).

to dissimilar climates or minimize distance travelled; 37% of source pixels had different destination locations when comparing the MED and ED approaches; of these pixels, the locations differed on average (geometric mean) by 236 km. MCE values were strongly right skewed; 71% of the continent exhibited no exposure (MCE = 0) despite having widely varying velocity$_{MED}$. For cells in which MCE > 0, the median value was 147 °C (3.0–8,005 °C). In addition to the results presented herein, which are based on gridded temperature data at 5 km resolution, we also assessed the sensitivity of velocity$_{MED}$, velocity$_{ED}$ and MCE to the resolution of input climate data, the width of temperature bins for defining a climate analogue, and cost penalties associated with the resistance surfaces used in least-cost modelling (see Methods section, Supplementary Fig. 1, Supplementary Fig. 2, and Supplementary Table 1).

There is considerable regional variability in velocity$_{MED}$ and MCE (summarized in Supplementary Table 2) driven by the rate of temperature change, regional landforms, proximity to water and local physiography. Velocity$_{MED}$ tended to increase with latitude due to greater projected warming, was lower in areas of complex terrain, and was higher in flat areas and regions surrounded by large water bodies such as the Great Lakes (Fig. 2, Supplementary Fig. 3). In contrast, MCE values were generally 0 in flat regions with high velocities (the Great Plains and Boreal region) and were often > 0 in topographically complex areas along the major mountain cordillera of the continent (areas with generally low velocity; Fig. 3). Areas with large MCE values often had climate trajectories that required southward movements (for example, northward oriented peninsulas) or traversed relatively warm valleys (mountainous regions; Figs 1b and 3, Supplementary Fig. 3). The ratio between velocity$_{MED}$ and velocity$_{ED}$ was also higher in mountainous regions and on northward oriented peninsulas (Fig. 2). Velocity$_{MED}$ and MCE are only weakly correlated (Spearman's rank $r = 0.37$); for MCE values > 0 this correlation increases ($r = 0.81$), although there is substantial variation in velocity$_{MED}$ across all MCE values (Fig. 4).

## Discussion

Velocity and MCE describe complementary facets of exposure: the length of potential migration paths and climatic resistance to movement along these paths through time. Our estimates of velocity$_{MED}$ and velocity$_{ED}$ mirror existing studies which suggest that mountainous regions have relatively low climate change

exposure (longer climate residence times)[3,10,12,14]. However, this interpretation assumes climatic resistance to movement does not vary in space. We show that in areas of complex terrain (for example, the cordillera of North America), MCE can be high even for short climate trajectories because they often traverse climatically heterogeneous landscapes (Figs 1b and 3). In this context, spatial climate heterogeneity acts to increase resistance to movement, thus reducing climatic connectivity between a source and destination pixel. In contrast, long climate trajectories over much of the continent pass through climatically homogenous environments with little to no resistance. Under these conditions, velocity is large but MCE is zero (that is, much of the Great Plains and Canadian Boreal region; Fig. 3).

Climate trajectories based on ED and MED define two end-members of a spectrum of potential trajectories that are contingent on a sites' physiographic setting and the traits of species that occupy the site. In flat regions, for example, there is little difference between these end-members; velocity$_{MED}$ and velocity$_{ED}$ show similar values (Fig. 2). In contrast, velocity$_{MED}$ and velocity$_{ED}$ diverge most notably in areas of complex terrain (Fig. 1a and 2). Furthermore, MCE represents the minimum exposure to dissimilar climate along a potential migration path. The actual climate change exposure of individual organisms will also depend on the traits of the species that occupy the site[21,22]. Vagile organisms may be able to keep pace with large climate velocities, whereas sessile organisms may fall behind changing climate[23], resulting in increased exposure as climate change outpaces movements. Moreover, dispersal traits of organisms will likely affect migration routes. For example, volant species such as birds may follow straight line paths that are well approximated by velocity$_{ED}$. Thermal tolerances may also influence whether an organism tends to follow straight line paths or paths that minimize exposure to dissimilar climates. For instance, thermal specialists (for example, ectotherms from the tropics;) are more likely to minimize their exposure to dissimilar climate (approximated by velocity$_{MED}$) as compared with thermal generalists (for example, endotherms from high latitudes)[22,24,25]. Similarly, the narrow reproductive niche[26–28] of many plant species will likely require longer migratory paths than those described using velocity$_{ED}$.

The bivariate distribution of velocity$_{MED}$ and MCE provides a logical framework for assessing exposure as a function of species dispersal capacity and thermal tolerance (Fig. 4a). We present a simple classification of the bivariate distribution of velocity$_{MED}$ and MCE as four groups (Fig. 4b): low velocity, low MCE; low

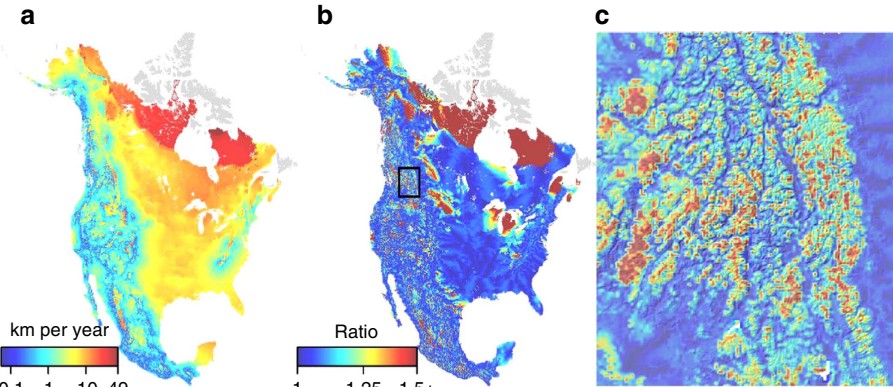

**Figure 2 | Climate velocity for North America for the interval 1995 to 2085.** Climate velocity based on minimum exposure distance (velocity$_{MED}$) is presented in **a**. The ratio between velocity$_{MED}$ and velocity based on Euclidean distance (velocity$_{ED}$) is presented in **b**. An inset (black box) of **b** resampled to a resolution of 1 km to improve visual interpretation is presented in **c**. Areas in grey represent pixels whose source or destination pixel is located on an island (see Methods section).

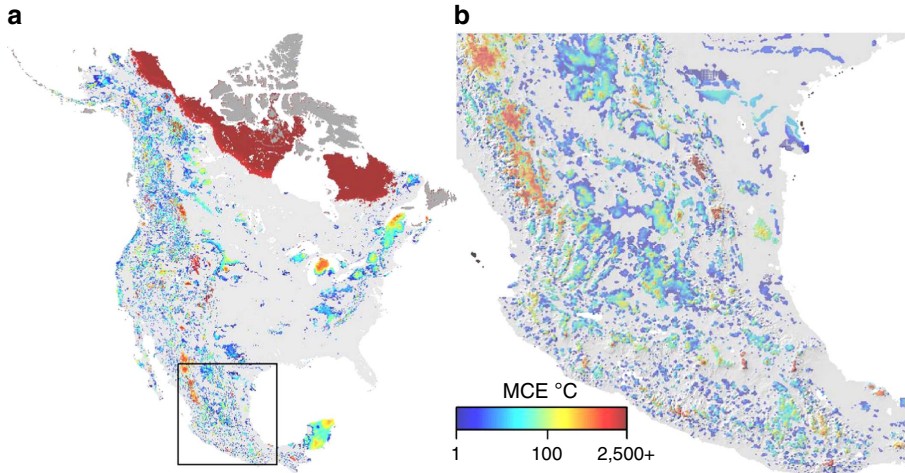

**Figure 3 | Minimum cumulative exposure.** MCE for North America for the interval 1995 to 2085 is presented in **a**. An inset (black box) resampled to a resolution of 1 km to improve visual interpretation is presented in **b**. MCE values > 0 are shown in colour; MCE = 0 is shown in light gray. MCE values for pixels on islands and pixels whose future climate analogues are on islands are shown in dark grey (see Methods section).

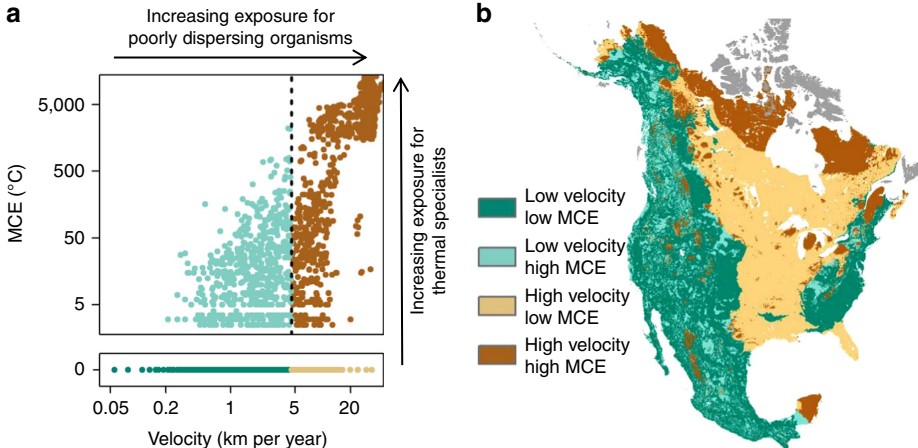

**Figure 4 | Classification of velocity and minimum cumulative exposure.** The relationship between velocity$_{MED}$ and minimum cumulative exposure (MCE) for a random sample of 5000 pixels from North America is presented in **a**. Velocity$_{MED}$ and MCE values > 0 are plotted on log-log axis. MCE values = 0 (71% of pixels) are plotted in lower panel and are considered low; MCE > 0 are considered high. The dashed vertical line represents the median velocity$_{MED}$ which is used to delineate between low and high velocity classes. Map showing the distribution of the four classes based on the bivariate distribution of MCE and velocity$_{MED}$ is presented in **b**. Dot colour in **a** corresponds to map color in **b**. Areas in grey represent pixels whose source or destination pixel is located on an island (see Methods section).

velocity, high MCE; high velocity, low MCE; and high velocity, high MCE. We use the terms low and high to simply qualify the range of our data and do not suggest that the specific thresholds we use (see methods) have biological relevance. Nevertheless, areas with high velocity present increased risk of exposure to poorly dispersing species. Areas of high MCE present increased risk for thermal specialists. Of particular concern are areas with both high velocity and MCE (for example, mountain tops, peninsulas). These areas are climatically isolated and increase the vulnerability of local populations to climate change. Specific examples include the Great Lakes, Yucatan Moist Forests, and the Northern Appalachian/Acadian ecoregions (Supplementary Table 2, Supplementary Fig. 4).

The use of climate velocity and MCE should be done in a manner that is cognizant of the tension between species-specific approaches, which have greater data needs but account for individual species responses to climate change, and coarse-filter approaches that are applied independently of species-specific data,

such as the metrics presented here. Distance-based velocity implicitly assumes negligible thermal tolerances or adaptive capacity of local populations and thus represents an upper limit for migration requirements for climate-sensitive biota[11]. In contrast, sites comprised of organisms with more broadly defined climate tolerances will have lower MCE and velocity values because more of the neighbouring landscape will be suitable for these organisms. This is demonstrated in Supplementary Table 1; mean velocity and MCE decrease as climate bin width increases. Fine-filter approaches that model potential migration paths for individual species have also been developed. For example, circuit theory has been applied to modelling faunal movement routes that track climate changes[29]. In addition, both distance- and gradient-based approaches for calculating velocity have been adapted to account for individual species climate tolerances; so-called biotic velocity estimates the direction and rate at which a given species must move to track its climatic niche[11,30]. Biotic velocity suffers from the same limitations we identify here;

it may not reflect the extent to which potential migration routes expose species to dissimilar climate. Nonetheless, the methods we employ can be applied to the calculation of biotic velocity and for estimating species-specific MCE.

Velocity$_{MED}$, velocity$_{ED}$ and MCE are also sensitive to the resolution of input data and to input parameters used in their calculation. Velocity$_{MED}$, velocity$_{ED}$ and MCE increase as data resolution is coarsened whereas they decline as the bin width for defining a climate analogue is broadened (Supplementary Table 1, Supplementary Fig. 1, Supplementary Fig. 2). These results are consistent with previous studies that have examined the scale sensitivity of velocity calculations[4,10] and suggest that comparisons between studies and comparisons of species dispersal rates against climate velocities (for example, ref. 23) must be interpreted with explicit consideration to scale of analysis.

Mountains support roughly a quarter of the globe's terrestrial biodiversity, contain 32% of protected areas and nearly half of the world's biodiversity hotspots[31]. In light of this, our results raise a logical question of how to reconcile the notion of mountains as refugia under changing climate versus mountains as areas of high climatic resistance to movement. Climate change adaptation strategies are often predicated on the former notion which assumes that spatial variability in climate may allow for short distance dispersal to ameliorate climate change impacts[14–16,32]. This paradigm underlies approaches for reserve design[33,34] and methods for identifying microrefugia[9,16,35,36]. Nonetheless, an important and sometimes overlooked consideration is whether sites are accessible to migrating organisms[37]. Accessibility is partly a function of climatic connectivity to a given site[4]. For example, velocity$_{MED}$ and MCE calculated for 'reverse' climate trajectories (sensu 'reverse velocity'[4,11]) provide two measures of climatic accessibility (Fig. 5). Accessibility will also depend on physical barriers to movement, habitat fragmentation, land use and land cover change, among other factors. Temporal scale is another important consideration when trying to reconcile

these two viewpoints. Velocity$_{MED}$ and MCE depend in part on the magnitude of climate shifts (a function of time) relative to the range or extent of the local spatial gradient in climate. Over short time intervals (or in places with sharp climate gradients) local dispersal, within the range of a local monotonic spatial climate gradient, may allow an organism to keep pace with climate change without exposure to dissimilar climates (for example, simple upslope movement; Fig. 1b, path 1). However when the magnitude of climate change exceeds the extent of the local spatial gradient in climate, MCE (and potentially MED) will increase sharply as organisms are required to traverse unsuitable climates en route to a future climate analogue (Fig. 1b, path 2). This implies that at certain spatial and temporal scales, montane sites will act as climate refugia. Beyond these spatio-temporal domains, these sites will provide only temporary holdout habitat (sensu[36]).

Climate velocity estimates (whether based on MED or ED) for montane regions suggest limited exposure and relatively short migration paths under climate change. However, we show that distance (and velocity) is an imperfect measure of climate connectivity given that it does not account for spatial variability in climatic resistance to movement (that is, MCE). MCE can be high in mountainous regions, which implies that these areas are more climatically isolated than has been previously reported. Conversely, flat regions of the continent have high velocities but negligible climatic resistance to movement. These regions may have greater climatic connectivity to their future climate analogues than is currently appreciated. Consequently, a more nuanced assessment of climate exposure is warranted, one which explicitly accounts for climatic resistance to movement. Hence, we advocate for further integration between 'climate informed' connectivity modelling approaches and the development of climate change exposure metrics. This integration will more realistically describe multiple facets of exposure to biota from ongoing climate change.

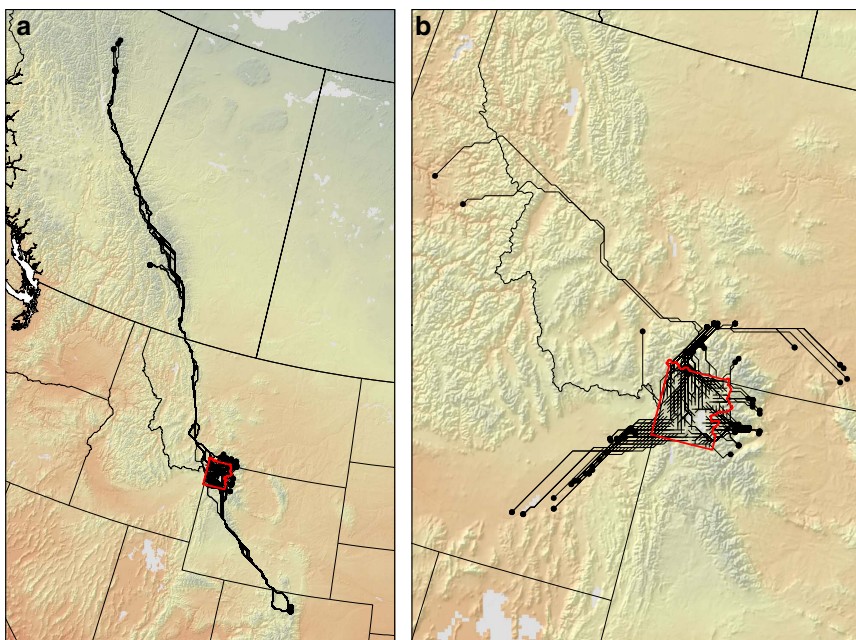

**Figure 5 | Climate trajectories for Yellowstone National Park USA.** Forward (**a**) and reverse (**b**) climate trajectories for Yellowstone National Park (red polygon) for the period 1995–2085. Forward trajectories depict routes that minimize the cumulative exposure (MCE) to dissimilar isotherms along paths between a source pixel and its future climate analogue (dot). Forward trajectories (sensu 'forward velocity'[4,11]) characterize the potential exposure of organisms to climate change. Reverse climate trajectories depict paths with the lowest MCE between a future cell and an analogue under current climate conditions (dot). Reverse trajectories (sensu 'reverse velocity'[4,11]) characterize how climatically accessible a site is to colonization from other sites. Climate trajectories (forward and reverse) can be characterized by minimum exposure distance (MED), velocity$_{MED}$ (length/time), and MCE.

## Methods

**Climate data.** We generated estimates of velocity and MCE over the interval (1981–2010) and (2071–2100) using mean annual temperature data described by (ref. 38). Projections of future mean annual temperature are based on an ensemble of 15 CMIP5 GCMs under the RCP 8.5 scenario. These data came rounded to the tenth of a degree and had an original spatial resolution of 1 km, which we resampled to a 5 km resolution.

For each 5 km pixel in North America, we identified all pixels with a climate analogue in the future time period. We define the climate analogue as any pixel that is ±0.25 °C from the pixel of interest, effectively setting a bin width of 0.5 °C. We then used a least-cost algorithm[39] to identify the trajectory (potential migration routes) that minimizes exposure to dissimilar climates between each source and destination pixel with a matching future climate analogue. We developed cost surfaces (one for each 0.1 °C temperature increment) in a manner that accounts for spatio-temporal changes in climate, allowing isotherms to be traced over landscapes and through time.

**Least-cost modelling.** The overall workflow for our approach to least-cost modelling is presented in Fig. 6. To account for a changing climate, we first interpolated mean annual temperature grids to represent incremental changes in temperature over the 90 year time period. These interpolated temperature grids ($n = 31$) represent mean annual temperature on about three year increments between the start (1995) and end date (2085) and assume a linear trend in changes in temperature over this time period (Fig. 6a). Next, we created intermediate cost surfaces (Fig. 6b,c) for each of these interpolated temperature grids such that costs were proportional to the level of dissimilarity between each 0.1 °C increment of

interest (in 1995) and all other pixels based on the following equation:

$$\text{cost}_{(i)} = 1 + \left(p \times \left|T - T_{(i)}\right|\right) \qquad (1)$$

where $\text{cost}_i$ is the cost assigned to pixel $i$, $P$ is a dissimilarity penalty ($P = 2$ dimensionless units per °C in this study such that $\text{cost}_i$ increases by 1 for each 0.5 °C in climate dissimilarity), $T$ is the 0.1 °C temperature increment of interest, and $T_i$ is the temperature of pixel $i$; a value of one is added because least-cost approaches do not allow costs of zero and it ensures that $\text{cost}_i =$ distance when trajectories pass through pixels with an analogous climate. We then computed, among all intermediate cost surfaces, the minimum cost for each pixel (Fig. 6b–d); this minimum cost is used to generate the final cost surfaces (Fig. 6d). Cost surfaces were developed for each 0.1 °C temperature increment ($n = 567$ cost surfaces). For each increment, climate analogues were defined based on bins that were 0.5 °C wide so that matching pixels (±0.25 °C) were assigned a cost of one and cost values increased with the level of dissimilarity as integers (for example, 5.3 °C ± 0.25–0.75 °C was assigned a cost of two; see Fig. 6, Table 1). We evaluated every 0.1 °C increment, yet used a bin width of 0.5 °C to define climate analogues. This was intended to reduce boundary effects; that is, we wanted to ensure that pixels with temperatures of 4.9 and 5.1 °C were considered climate analogues as opposed to being treated as separate bins (Table 1). In the final cost surfaces, lakes and ocean were arbitrarily assigned a cost value of 5,000 to heavily penalize open water, thereby forcing trajectories to avoid water when possible (Supplementary Fig. 3). Excluding water, cost values generally ranged from 1 to ~100, reflecting the ~50 °C difference in mean annual temperature across North America.

Accounting for temporal changes in climate (Fig. 6a–d) was necessary for two reasons. First, had we based our cost surface on the mean annual temperature in

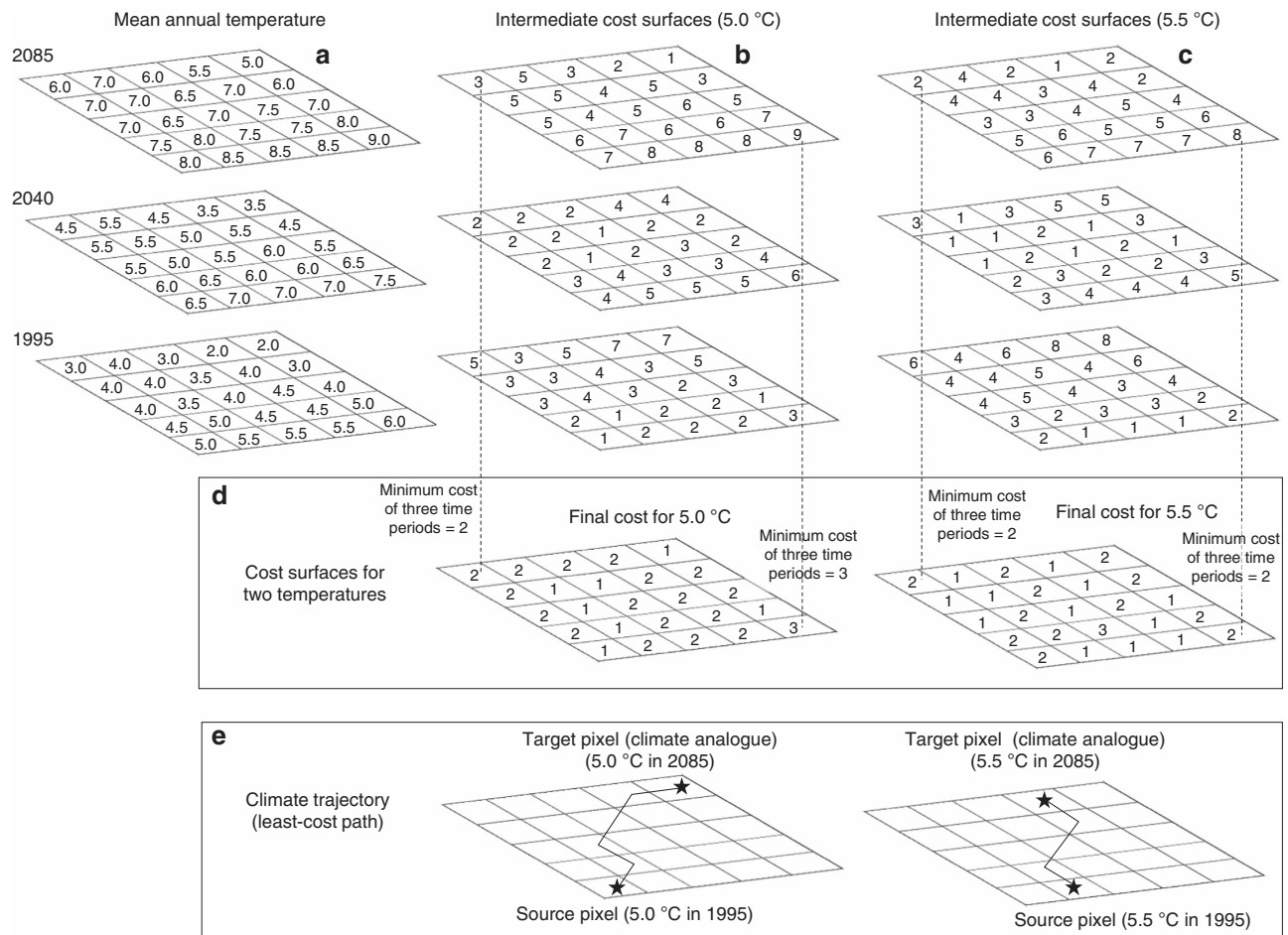

**Figure 6 | Workflow for least-cost modelling.** Illustration of how final cost surfaces were generated for two temperature increments (5.0 and 5.5 °C) and two examples of climate trajectories for the interval between 1995 and 2085. To account for changing temperature fields through time, we produced intermediate cost surfaces between the start and end years (**a**). Here we show a single intermediate temperature field for simplicity; however our analysis utilized 31 intermediate temperature surfaces which were linearly interpolated between 1995 and 2085. Cost surfaces were generated based on equation 1 for the 5.0 °C isotherm (**b**) and 5.5 °C isotherm (**c**) (see Methods section). For each pixel, the minimum cost over all time periods is retained for the final cost surface (**d**), which is then used to generate climate trajectories using a least-cost algorithm (**e**). Note that the example climate trajectories have the lowest accumulated cost compared to any other potential trajectories. This approach accounts for a changing climate by allowing trajectories to follow the climate as it warms. Cost surfaces were generated in this fashion for each 0.1 °C increment (Table 1).

**Table 1 | Examples of how climate analogues and costs were defined in the study.**

| Temperature of source pixel | Cost = 1 (climate analogue) | Cost = 2 | Cost = 3 | Cost = 4 |
|---|---|---|---|---|
| 5.0 °C | 5.0 °C ± 0.25<br>●4.75–5.25 °C | 5.0 °C ± (0.25–0.75)<br>●4.25–4.75 °C<br>●5.25–5.75 °C | 5.0 °C ± (0.75–1.25)<br>●3.75–4.25 °C<br>●5.75–6.25 °C | 5.0 °C ± (1.25–1.75)<br>●3.25–3.75 °C<br>●6.25–6.75 °C |
| 5.1 °C | 5.1 °C ± 0.25<br>●4.85–5.35 °C | 5.1 °C ± (0.25–0.75)<br>●4.35–4.85 °C<br>●5.35–5.85 °C | 5.1 °C ± (0.75–1.25)<br>●3.85–4.35 °C<br>●5.85–6.35 °C | 5.1 °C ± (1.25–1.75)<br>●3.35–3.85 °C<br>●6.35–6.85 °C |
| 5.2 °C | 5.2 °C ± 0.25<br>●4.95–5.45 °C | 5.2 °C ± (0.25–0.75)<br>●4.45–4.95 °C<br>●5.45–5.95 °C | 5.2 °C ± (0.75–1.25)<br>●3.95–4.45 °C<br>●5.95–6.45 °C | 5.2 °C ± (1.25–1.75)<br>●3.45–3.95 °C<br>●6.45–6.95 °C |

Examples show increasing temperature dissimilarities for source pixels with 5.0, 5.1 and 5.2 °C.

one time period (for example, 1995), trajectories would reflect the minimum exposure in that time period only and would not reflect the optimum path based on a changing climate. The second reason is one of accounting: MCE would always be > 0 if we did not base the final cost surfaces on the interpolated temperature grids. For example, had we based cost on the mean annual temperature in 1995, the cost associated with the destination location, which by definition is a climate analogue and should have a cost equal to one (equation 1), would be high because the cost would reflect 1995 climate and not 2085 climate (compare top and bottom panels in Fig. 6b,c). Consequently, our approach accounts for a changing climate by allowing trajectories to follow the climate as it warms and acknowledges that paths between source and destination locations with no intervening topographic variation should not traverse dissimilar climates and will have an MCE = 0 (for example, simple northward migration in flat terrain).

For each source pixel in each temperature increment, we used the costDistance function of the gdistance package[40] in the R statistical environment to calculate the accumulated cost to all pixels with the corresponding future climate analogue. We then identified the destination pixel with the least-accumulated cost and used the shortestPath function of the gdistance package to delineate the trajectory from source to destination pixel (Fig. 6e). By definition, trajectories delineated with these methods minimize exposure to dissimilar climates (as defined by our cost surfaces) and the length (km) of these trajectories is termed the MED. We subsequently calculated velocity$_{MED}$ (km year$^{-1}$) as:

$$\text{velocity}_{MED} = \frac{\text{MED}}{\text{time}} \qquad (2)$$

where *time* is the elapsed time between current and future time periods (*time* = 90 years in this study). MCE was calculated as:

$$\text{MCE} = \frac{\sum_s^d (\text{cost}_i \times l_i) - \text{MED}}{p} \qquad (3)$$

The term $\sum_s^d (\text{cost}_i \times l_i)$ is the least-accumulated cost, a standard output of least-cost methods where $\text{cost}_i$ is the cost assigned by equation 1 and is summed from $s$, the source pixel to $d$, the destination pixel, $l_i$ is the length (km) of the trajectory though pixel $i$ (this acknowledges that a diagonal trajectory through pixel $i$ is longer than the horizontal or vertical equivalent; Fig. 6d), MED is the minimum exposure distance (which we subtract from total accumulated cost because it implicitly includes path distance; this is a result of assigning a value of 1 to all pixels with the same climate analogue; equation 1), and $p$ is a climate dissimilarity penalty (as described in equation 1). In summary, MCE is a cumulative tally of climate dissimilarities that were encountered along each path. For example, MCE = 2.5 °C indicates that the path, regardless of its total length, traversed 5 km (that is, one pixel; the resolution of our climate data) that was ± 0.5 °C from the pixel of interest; MCE = 5.0 °C indicates that the path, regardless of its length, traversed 10 km (that is, two pixels) that was ± 0.5 °C from the pixel of interest OR traversed 5 km (one pixel) that was ± 1.0 °C from the pixel of interest. For comparative purposes, we also identified the ED nearest neighbour and ED-based velocity (cf., ref. 4) using the same climate analogue and binning approach described above.

We calculated velocity$_{MED}$, velocity$_{ED}$, the ratio of velocity$_{MED}$ to velocity$_{ED}$ and MCE for North America and produced corresponding maps. We excluded pixels from these maps (and all statistical analysis) where the source or destination pixel was located on an island; MCE values are not interpretable for these areas because we assigned large costs to water to ensure that trajectories circumvented water when possible (Supplementary Fig. 3).

To help visualize spatial patterns in exposure, we classified pixels within North America using a simple four-category scheme based on the bivariate distribution of velocity$_{MED}$ and MCE. Given that 71% of the MCE values were 0, we chose MCE > 0 to delineate 'high' values and MCE = 0 to delineate 'low' values. For velocity$_{MED}$, we used the median as the boundary between 'high' and 'low' values.

**Sensitivity analysis.** To evaluate the sensitivity of our results to input parameters and resolution of climate data, we conducted a sensitivity analyses by altering (1)

the resolution of the climate data from 2 to 20 km, (2) the bin width for defining a climate analogue and (3) the cost penalty associated with dissimilar climates (equation 1) for a subset of N. America. Specifically, we used the state of Montana, USA as a subset as it has both mountainous and flat terrain. For each permutation, we report the geometric mean, and the 5th, 50th and 95th percentiles for velocity$_{MED}$, velocity$_{ED}$, the ratio between velocity$_{MED}$ and velocity$_{ED}$, and MCE (Supplementary Table 1, Supplementary Fig. 1, Supplementary Fig. 2).

**Code availability.** R code for implementing the methods is available at: http://adaptwest.databasin.org/pages/adaptwest-velocitymed

**Data availability.** Outputs from the analysis are available at: http://adaptwest.databasin.org/pages/adaptwest-velocitymed

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

## Acknowledgements

S.Z.D. was supported by the National Science Foundation (DEB 1145985; BCS 1461576) and USFS Rocky Mountain Research Station (Agreement number: 15-JV-11221639-119). We thank Brady Allred for providing computing resources, and John Abatzoglou and three anonymous reviewers for constructive comments on previous drafts of this manuscript.

## Author contributions

S.Z.D. and S.A.P. contributed equally to the design, analysis, and writing of this paper.

## Additional information

**Competing financial interests:** The authors declare no competing financial interests.

