## [Peer Review File · Nature Communications]

Reviewers' comments:

Reviewer #1 (Remarks to the Author):

This is a thought-provoking paper on the constraints imposed by local spatial variability in average temperature on species whose distributions may track temperature in a changing climate. By extending the idea of climate velocity as an analogy of the processes that underpin change in species' distributions, they show effectively that places with lots of local variability in average temperature, such as mountainous areas, present greater resistance to range shifts than places where temperatures are more evenly distributed. The two novel metrics presented, the minimum exposure distance (MED) and minimum cumulative exposure (MCE) are well explained and are useful. MED gives a more realistic (and larger) estimate of the distance to the nearest climate analog than Euclidean distance (ED) by using a least cost path approach, while MCE is a new measure of how difficult it may be to traverse unfavourable conditions at intermediate points between current places and their future climate analogs. The methods explain the production of these metrics very well, though it is clear that there is considerably more computation involved than in the original climate velocity or trajectory approaches, but with the benefit of incorporating much greater realism by not requiring shifts only along gradients of decreasing temperature for velocities in warming conditions.

I found the account of how species distributions might follow the proposed least-cost trajectory paths confused (line 127 and thereabouts "An organism follows..."). It must be remembered that the climate trajectory is a simulation of dynamics at geographical range edges and that expansion and contraction in real distributions involves many ecological and even physiological processes. At contracting range edges, no trajectory tracking is even necessary: populations may simply become isolated and then disappear as temperatures exceed thresholds for survival or performance necessary for population maintenance. So the discussion of species dispersal traits applies only to expanding range edges. The thermal tolerance range of a species is probably irrelevant in terms of the species' ability to tolerate intermediate dissimilar climates (lines 128-129) since at the edge of the species range by definition the species can only just tolerate the existing conditions and all other climates are outside that envelope. Mobility and the ability to colonise new areas are key traits of course. Likewise line 139 where areas of high MCE are presented as problematic for thermal specialists, it seems unfounded to link the species ability to transcend unfavourable areas of climate at the edge of their range to the width of their realised thermal niche.

The discussion of the relative merits of this non-specific approach and species-specific models, alongside other approaches such as 'biotic' velocity, is well placed.

Considering the previous work of these authors, it is surprising that the issue of scale is not addressed. In their 2013 paper they show nicely how strong an effect scale has on magnitude and direction of climate velocity. Here, it is the magnitude of the local scale variability in temperature that drives the minimum cumulative exposure and the difference between Euclidean distance derived and MED velocity. Above a certain scale these important local barriers imposed by unfavourable intervening climates will be averaged out, and MED, ED and velocity based on gradients alone will converge towards similar values (as the paper shows to be the case already in

areas where local scale variability is minimal).

Reviewer #2 (Remarks to the Author):

This manuscript is novel and of broad interest to those working in areas related to assessing climate change impacts on species. The basic idea behind the approach is not entirely novel, but never has been implemented in the past to my knowledge, due in part to computational feasibility issues. The authors also present the approach in a more coherent conceptual framework (using the metrics MCE and MED) than has been done in previous work by others. However, the authors should add more context regarding earlier work such as by Lawler et al. 2013 (Lawler JJ, Ruesch AS, Olden JD, Mcrae BH (2013) Projected climate-driven faunal movement routes. *Ecol Lett*, 16, 1014-1022.), who identified climatically-constrained movement linkages using current flow rather than least-cost path. Dobrowski and Parks' approach could be in principle generalized to use current flow, although this might be computationally challenging.

The concept of climatic exposure, which is primary focus of this work, is inherently fairly broad and vague, which creates some challenges concerning the manuscript's statements that a certain metric is a better representation of exposure. The key theme of paper that should be emphasized in my view is that "velocity and MCE depict complementary facets of exposure" (line 21), along with examples of how would they be used to complement each other (e.g. in Figure 4), rather than the superiority of one metric over another. It is true that Euclidean distance is "an imperfect measure of exposure", but we are really talking about 2 models (Euclidean and least-cost distance) which are both imperfect but which when considered together, may provide complementary information. In part the respective relevance of the two metrics will depend on the dispersal ability of taxa of interest (e.g. birds vs amphibians), a point mentioned in the introduction (lines 52-64). But the point that both metrics are imperfect surrogates for connectivity could be clarified.

A key additional area that needs clarification is the contrasting interpretations of the three metrics analyzed: Euclidean distance (henceforth ED, the basis for previous distance-based climatic velocity metrics), MCE and MED. It wasn't until I had reviewed the supplementary information that I understood the contrast between MCE and MED. As it stands now, the main manuscript confuses readers in that the authors contrast both MCE and MED with ED before they clearly compare MCE and MED with each other. And some general statements about the contrasts of their newly-developed metrics (MED, MCE) with ED only apply to MCE and not to MED, or vice versa. For example, results (lines 89-104) contrasts ED with MCE, but it seems that the most relevant comparison here would be ED vs MED, and this comparison should be added. To some extent, the authors are focusing on the more dramatic contrast between ED and MCE, without acknowledging that to some extent this is apples and oranges, and that ED and MED provide the more relevant comparison in many cases.

This extends to the discussion, for example in lines 161-163 where the overarching conclusions confuses MCE and MED to some extent: "Our results are consistent with existing work; areas of complex terrain have shorter potential migration paths. However, we find that increasing spatial variability in climate comes with the cost of increasing climatic resistance to movement. Climate

change exposure metrics rarely consider these threats." If I am interpreting this correctly, this text is stating: "ED is similar to MED in montane areas. MCE contrasts with ED in montane areas." But upon first read, this interpretation may not be evident to the reader.

The key conceptual conclusion concerning interpretation and practical relevance of the results should be clarified and given more context. Basically, this is the question of how to reconcile the concept of mountains as climatic refugia vs mountains as areas of high climatic dissimilarity (MCE). This is in part a scale issue, in that it is dependent on magnitude of climate shift relative to extent of local climatic spatial gradient. It would be possible for a montane climatic gradient to show a MCE of zero if local dispersal (within the extent of a local unidirectional climate gradient) allowed an organism to keep pace with climate. But if/when the magnitude of climatic change exceeded the magnitude of the local gradient, then MCE (and potentially MED) would increase sharply as organisms were required to traverse heterogeneous montane climates to reach suitable future climate space.

So this implies that the common hypothesis that mountains are potential climate refugia remains true at certain spatial and temporal scales. The value of this paper is in part demonstrating/quantifying that this hypothesis is false beyond certain scales and beyond a certain magnitude of climate change. This results in mountains providing only temporary "holdout" habitat (sensu Hannah et al. 2014 Fine-grain modeling of species' response to climate change: holdouts, stepping-stones, and microrefugia. *Trends Ecol Evol.*). I think the discussion should be strengthened by treating this issue at greater length and/or more clearly.

Two additional sensitivity analyses seem necessary: The statement (line 85) that 74% of the continent exhibited zero MCE is likely contingent on the width of the climate bin. Similar results can be achieved using Euclidean distance-based velocity if bins are broad (see treatment in Hamman et al. 2014 and Carroll et al. 2015 Figure S1). A sensitivity analysis of the mean or distribution of MCE vs bin width is necessary here.

Also, MED effectively is a model, like all least-cost path models, which attempts to minimize some joint function of climatic dissimilarity and geographic distance. The weight given to the respective units of climatic dissimilarity (i.e., the unit cost per degree of climate change) and geographic distance is somewhat arbitrary and influenced by resolution. The equation for MCE accounts for the effects of resolution in the denominator, but the function for MED does not, so some analysis of whether results differ under different resolutions (or using different units for distance and climate dissimilarity) would be useful.

Reviewer #3 (Remarks to the Author):

I liked this a lot and it has made me re-think some of my own current work. The conclusions are strikingly novel - opposite to the most often quoted ones - and appear to be robust. The figures and SM are very helpful, but I would like to have seen a little more explanation of the MED and MCE in the Methods section of the main text. Currently every reader has to read the SM to understand anything. I would also like to see a justification for the 5 km pixel size - this seems low for

mountainous topography, except for species that can disperse that far over hostile terrain - and the 0.5 degree C bin width for climate analogs, which seems arbitrary (although I have no better suggestion). Optionally, I think it would be useful to discuss briefly what MED and MCE means for, say, plants, large mammals, and birds.

Response to reviewers

Reviewer #1:

This is a thought-provoking paper on the constraints imposed by local spatial variability in average temperature on species whose distributions may track temperature in a changing climate. By extending the idea of climate velocity as an analogy of the processes that underpin change in species' distributions, they show effectively that places with lots of local variability in average temperature, such as mountainous areas, present greater resistance to range shifts than places where temperatures are more evenly distributed. The two novel metrics presented, the minimum exposure distance (MED) and minimum cumulative exposure (MCE) are well explained and are useful. MED gives a more realistic (and larger) estimate of the distance to the nearest climate analog than Euclidean distance (ED) by using a least cost path approach, while MCE is a new measure of how difficult it may be to traverse unfavourable conditions at intermediate points between current places and their future climate analogs. The methods explain the production of these metrics very well, though it is clear that there is considerably more computation involved than in the original climate velocity or trajectory approaches, but with the benefit of incorporating much greater realism by not requiring shifts only along gradients of decreasing temperature for velocities in warming conditions.

Thank you for the feedback.

I found the account of how species distributions might follow the proposed least-cost trajectory paths confused (line 127 and thereabouts "An organism follows..."). It must be remembered that the climate trajectory is a simulation of dynamics at geographical range edges and that expansion and contraction in real distributions involves many ecological and even physiological processes. At contracting range edges, no trajectory tracking is even necessary: populations may simply become isolated and then disappear as temperatures exceed thresholds for survival or performance necessary for population maintenance. So the discussion of species dispersal traits applies only to expanding range edges.

The thermal tolerance range of a species is probably irrelevant in terms of the species' ability to tolerate intermediate dissimilar climates (lines 128-129) since at the edge of the species range by definition the species can only just tolerate the existing conditions and all other climates are outside that envelope. Mobility and the ability to colonise new areas are key traits of course. Likewise line 139 where areas of high MCE are presented as problematic for thermal specialists, it seems unfounded to link the species ability to transcend unfavourable areas of climate at the edge of their range to the width of their realised thermal niche.

We thank the reviewer for their comments. First we would like to respectfully disagree with the notion that movement of organisms along climate trajectories is only relevant at the leading edge of species distributions. Indeed, the notion that at trailing edges existing populations may simply contract and disappear is true. However, this does not preclude niche tracking by organisms via dispersal and recruitment. For example, there is evidence that recruitment at lower treeline for trailing edge populations of western US trees is occurring at higher elevations and in more mesic settings (Dobrowski et al. 2015). As the reviewer suggests, adult trees at these lower treeline sites will likely disappear due to drought and/or disturbance and will not be replaced by recruits. Velocity trajectories for these regions provide insights into where recruitment could occur (e.g. upslope) and the dispersal/migration rates that would be needed for these low elevation populations to keep pace with projected climate shifts. Of course the climate analogs for these low elevation forest sites may occur within the current range of the species but this doesn't preclude the utility of the approach as a coarse-filter exposure metric. Maybe more importantly than a discussion about niche tracking at leading vs. trailing edges, is the recognition that climate velocity and MCE should be cautiously interpreted for a given species and are more appropriately viewed as metrics of climate change exposure for a given site (lines 157-171). Through this lens, it should be noted that the trailing edge of one population is the leading edge of another. High velocities and high MCE suggest that a site (with its assemblage of organisms) are climatically isolated and thus have high potential exposure to climate change. The distinction between leading vs. trailing edge is more appropriately applied to a fine filter approach such as the use of bioclimatic velocity (lines 162-171). Similarly, our description of species traits including dispersal capacity and thermal tolerance should be viewed through this same coarse-filter lens. Sites comprised of organisms with narrow thermal tolerances will be more vulnerable if those sites have high values of MCE. For example, a climate trajectory for a site may traverse a region of dissimilar climates (e.g. a mountain range) en route to its climate analog. Species with broad thermal tolerances are more likely to be able to tolerate this movement compared to species with narrow thermal tolerances.

The discussion of the relative merits of this non-specific approach and species-specific models, alongside other approaches such as 'biotic' velocity, is well placed.

Thank you.

Considering the previous work of these authors, it is surprising that the issue of scale is not addressed. In their 2013 paper they show nicely how strong an effect scale has on magnitude and direction of climate velocity. Here, it is the magnitude of the local scale variability in temperature that drives the minimum cumulative exposure and the difference between Euclidean distance derived and MED velocity. Above a certain scale these important local barriers imposed by unfavourable intervening climates will be averaged out, and MED, ED and velocity based on gradients alone will converge towards similar values (as the paper shows to be the case already in areas where local scale variability is minimal).

Thank you for the comment and noting our previous work on this issue. We have now included a set of sensitivity analysis in the revised ms where we examine the influence of data resolution, climate bin width, and the cost penalty associated with dissimilar climates on $velocity_{MED}$, $velocity_{ED}$, and

MCE. Computational constraints make it prohibitive to conduct this sensitivity analysis across all of N. America. Consequently, we have run the analysis for the state of Montana as it contains both mountainous regions and plains. We have described the sensitivity analysis in the results and discussion sections (lines 173-180). Additionally, we have included figures S3 and S4 along with Table S2 in the supplemental summarizing our findings.

Reviewer #2:

This manuscript is novel and of broad interest to those working in areas related to assessing climate change impacts on species. The basic idea behind the approach is not entirely novel, but never has been implemented in the past to my knowledge, due in part to computational feasibility issues. The authors also present the approach in a more coherent conceptual framework (using the metrics MCE and MED) than has been done in previous work by others. However, the authors should add more context regarding earlier work such as by Lawler et al. 2013 (Lawler JJ, Ruesch AS, Olden JD, Mcrae BH (2013) Projected climate-driven faunal movement routes. *Ecol Lett*, 16, 1014-1022.), who identified climatically-constrained movement linkages using current flow rather than least-cost path. Dobrowski and Parks' approach could be in principle generalized to use current flow, although this might be computationally challenging.

Thank you for the comment and suggestions for citations to relevant earlier work. We have included the citation by Lawler et al. (2013) in the revised ms in the discussion (Line 165).

The concept of climatic exposure, which is primary focus of this work, is inherently fairly broad and vague, which creates some challenges concerning the manuscript's statements that a certain metric is a better representation of exposure. The key theme of paper that should be emphasized in my view is that "velocity and MCE depict complementary facets of exposure" (line 21), along with examples of how would they be used to complement each other (e.g. in Figure 4), rather than the superiority of one metric over another. It is true that Euclidean distance is "an imperfect measure of exposure", but we are really talking about 2 models (Euclidean and least-cost distance) which are both imperfect but which when considered together, may provide complementary information. In part the respective relevance of the two metrics will depend on the dispersal ability of taxa of interest (e.g. birds vs amphibians), a point mentioned in the introduction (lines 52-64). But the point that both metrics are imperfect surrogates for connectivity could be clarified.

Thank you for the comment. We have reviewed the ms with an eye to rewording language to emphasize the points made by the reviewer. For example, we have deleted lines 100-101 which as originally worded implied that MED provides a more accurate estimate of migration distance. Similarly, in the discussion, we have reworded lines 131-132. We agree with the reviewers point that both ED and MED bracket a spectrum of possible climate trajectories that will depend in part on the traits of species that occupy a site. To further emphasize this point we have included the following language in the revised ms (lines 125-129):

"Climate trajectories based on ED and MED define two end-members of a spectrum of potential climate trajectories that are contingent on a sites' physiographic setting and the traits of species that

occupy the site. In flat regions, there is little difference between these end-members; $velocity_{MED}$ and $velocity_{ED}$ show similar values (Fig.2). In contrast, $velocity_{MED}$ and $velocity_{ED}$ diverge most notably in areas of complex terrain (Fig 1a, Fig. 2).”

We would note that in the original ms we never explicitly state that ED is an ‘imperfect measure of exposure’ or that MED is superior to ED. Instead, we state (line 52) that ‘distance’ itself (regardless of whether it is based on ED or MED) is an incomplete description of connectivity because it lacks information about climatic resistance to movement, thus emphasizing the need for characterizing climatic resistance to movement.

A key additional area that needs clarification is the contrasting interpretations of the three metrics analyzed: Euclidean distance (henceforth ED, the basis for previous distance-based climatic velocity metrics), MCE and MED. It wasn't until I had reviewed the supplementary information that I understood the contrast between MCE and MED. As it stands now, the main manuscript confuses readers in that the authors contrast both MCE and MED with ED before they clearly compare MCE and MED with each other. And some general statements about the contrasts of their newly-developed metrics (MED, MCE) with ED only apply to MCE and not to MED, or vice versa. For example, results (lines 89-104) contrasts ED with MCE, but it seems that the most relevant comparison here would be ED vs MED, and this comparison should be added. To some extent, the authors are focusing on the more dramatic contrast between ED and MCE, without acknowledging that to some extent this is apples and oranges, and that ED and MED provide the more relevant comparison in many cases.

We posit that the confusion noted by the reviewer stems from our use of the term ‘velocity’ in the results section without consistently stating if velocity is calculated using MED or ED. Most of the results section (lines 89-104 original ms) refer to velocity based on MED which we have now made explicit using the terms $velocity_{MED}$ and $velocity_{ED}$. It is also important to note, that we do not contrast $velocity_{ED}$ at any point to MCE given that the comparison is not appropriate as the reviewer states. For example, the only quantitative comparison involving MCE is one in which we describe the correlation between MCE and $velocity_{MED}$ made in lines 101 and 102. Qualitative comparisons involving MCE are made against $velocity_{MED}$ (lines 89-104). Further, a careful read of our results will show that the only comparison using $velocity_{ED}$ is one in which we compare it to $velocity_{MED}$ on a percent basis (lines 80-81) and when describing the ratio of the two velocities in flat vs. mountainous regions (lines 98-99). We feel the latter comparison is warranted and informative.

This extends to the discussion, for example in lines 161-163 where the overarching conclusions confuses MCE and MED to some extent: "Our results are consistent with existing work; areas of complex terrain have shorter potential migration paths. However, we find that increasing spatial variability in climate comes with the cost of increasing climatic resistance to movement. Climate change exposure metrics rarely consider these threats." If I am interpreting this correctly, this text is stating: "ED is similar to MED in montane areas. MCE contrasts with ED in montane areas." But upon first read, this interpretation may not be evident to the reader.

The reviewers interpretation of lines 161-163 is not consistent with our intentions and reflects a lack

of clarity on our part. We are simply trying to state that velocity estimates (whether based on MED or ED) in mountain regions suggest limited climate change exposure and relatively short migration paths. However, these metrics do not account for the fact that areas with high spatial variability in climate also have high climatic resistance to movement (as measured by MCE). Thus, we are not trying to make an explicit comparison of ED with MCE. To address this ambiguity we have reworded this section as follows (lines 206-209 in revised ms):

“Climate velocity estimates (whether based on MED or ED) for montane regions suggest limited exposure and relatively short migration paths under climate change. However, we show that distance (and velocity) is an imperfect measure of climate connectivity given that it does not account for spatial variability in climatic resistance to movement (i.e. MCE).”

The key conceptual conclusion concerning interpretation and practical relevance of the results should be clarified and given more context. Basically, this is the question of how to reconcile the concept of mountains as climatic refugia vs mountains as areas of high climatic dissimilarity (MCE). This is in part a scale issue, in that it is dependent on magnitude of climate shift relative to extent of local climatic spatial gradient. It would be possible for a montane climatic gradient to show a MCE of zero if local dispersal (within the extent of a local unidirectional climate gradient) allowed an organism to keep pace with climate. But if/when the magnitude of climatic change exceeded the magnitude of the local gradient, then MCE (and potentially MED) would increase sharply as organisms were required to traverse heterogeneous montane climates to reach suitable future climate space. So this implies that the common hypothesis that mountains are potential climate refugia remains true at certain spatial and temporal scales. The value of this paper is in part demonstrating/quantifying that this hypothesis is false beyond certain scales and beyond a certain magnitude of climate change. This results in mountains providing only temporary "holdout" habitat (sensu Hannah et al. 2014 Fine-grain modeling of species' response to climate change: holdouts, stepping-stones, and microrefugia. *Trends Ecol Evol.*). I think the discussion should be strengthened by treating this issue at greater length and/or more clearly.

We thank the reviewer for a very insightful set of comments here. These are salient points that we have now included in an expanded discussion on this issue (lines 182-205 in revised ms). A portion of this text follows:

“Temporal scale is another important consideration when trying to reconcile these two viewpoints. $Velocity_{MED}$ and MCE depend in part on the magnitude of climate shifts (a function of time) relative to the range or extent of the local spatial gradient in climate. Over short time intervals (or in places with sharp climate gradients) local dispersal, within the range of a local monotonic spatial climate gradient, may allow an organism to keep pace with climate change without exposure to dissimilar climates (e.g. simple upslope movement; Fig. 1b, path 1). However when the magnitude of climate change exceeds the extent of the local spatial gradient in climate, MCE (and potentially MED) will increase sharply as organisms are required to traverse unsuitable climates en route to a future climate analog (Fig. 1b, path 2). This implies that at certain spatial and temporal scales,

montane sites will act as climate refugia. Beyond these spatio-temporal domains, these sites will provide only temporary 'holdout' habitat (*sensu*³⁶). "

Thank you for the helpful comment.

Two additional sensitivity analyses seem necessary: The statement (line 85) that 74% of the continent exhibited zero MCE is likely contingent on the width of the climate bin. Similar results can be achieved using Euclidean distance-based velocity if bins are broad (see treatment in Hamman et al. 2014 and Carroll et al. 2015 Figure S1). A sensitivity analysis of the mean or distribution of MCE vs bin width is necessary here.

Also, MED effectively is a model, like all least-cost path models, which attempts to minimize some joint function of climatic dissimilarity and geographic distance. The weight given to the respective units of climatic dissimilarity (i.e., the unit cost per degree of climate change) and geographic distance is somewhat arbitrary and influenced by resolution. The equation for MCE accounts for the effects of resolution in the denominator, but the function for MED does not, so some analysis of whether results differ under different resolutions (or using different units for distance and climate dissimilarity) would be useful.

We have now included a set of sensitivity analysis in the revised ms where we examine the influence of data resolution, climate bin width, cost penalty on $velocity_{MED}$, $velocity_{ED}$, and MCE. Computational constraints make it prohibitive to conduct this sensitivity analysis across all of N. America. Consequently, we have run the analysis for the state of Montana as it contains both mountainous regions and plains. We have described the sensitivity analysis in the methods, and discussion. Additionally, we have included figure S3 and S4 and Table S2 in the supplemental summarizing our findings.

Please note that, in conducting the sensitivity analysis we found that the resolution term in the denominator of the original MCE equation (eqn. 3) precluded us from making direct comparisons among resolutions. Consequently we updated the MCE equation (we now exclude the resolution term in the denominator), thereby allowing for a more consistent interpretation of the results across different resolutions. These changes are described further in the methods section (lines 292-311).

Reviewer #3:

I liked this a lot and it has made me re-think some of my own current work. The conclusions are strikingly novel - opposite to the most often quoted ones - and appear to be robust. The figures and SM are very helpful, but I would like to have seen a little more explanation of the MED and MCE in the Methods section of the main text. Currently every reader has to read the SM to understand anything.

Thank you for the comments. We also found the results novel in a way that challenged our previous

conceptions about the role of mountains as refugia. We agree with the reviewer about the need to move the methods into the body of the manuscript and have done so. This should aid in reading the ms.

I would also like to see a justification for the 5 km pixel size - this seems low for mountainous topography, except for species that can disperse that far over hostile terrain - and the 0.5 degree C bin width for climate analogs, which seems arbitrary (although I have no better suggestion).

Put simply, 5km was the finest resolution we could run for all of N. America given the computational requirements of the least-cost algorithm and our current access to computing resources. Presumably, this could be improved with access to cloud computing resources.

Optionally, I think it would be useful to discuss briefly what MED and MCE means for, say, plants, large mammals, and birds.

We currently include references to plants and mammals in the discussion when describing the interpretation of MED and MCE (lines 129-130). We have also included a reference to birds in the revision (lines 135-136).

Reviewers' comments:

Reviewer #1 (Remarks to the Author):

My comments here reflect only a divergence of opinion on the interpretation of the proposed metrics, rather than their implementation.

I contended that the trajectory following approach would be mostly if not wholly applicable to leading edges of ranges, and the counterargument does not entirely satisfy. Granted not all trailing edge populations will be adjacent to core range populations and that some travel would be needed for those population members to traverse unfavorable habitat. But fundamentally the processes of retreat differ from those of population advance, and unless we are talking about mobile organisms, this is the difference between local colonization (dispersal) and extinction (death, reproductive failure).

It is also an interesting idea to look at these metrics from the viewpoint of their utility as measures of exposure to climate change, rather than their ability to predict actual shifts in ranges. But that makes them much less useful and almost impossible to validate with any real data.

The focus of the metrics is on range edges where all organisms are on the brink of their existing tolerance irrespective of the width of their thermal niche. Traversing a range of climates that are outside the thermal niches is as likely to impact species with narrow ranges as those with broad ones. I respectfully disagree that thermal range width is relevant to the vulnerability to the effects of isolation (high MCE values). Communities comprised of narrow ranged species are vulnerable to climate change wherever they are, isolated or not.

Reviewer #2 (Remarks to the Author):

The authors have adequately addressed comments in the initial reviews, resulting in a very useful contribution to the field.

My sole remaining suggestion is that the authors provide a sample R script (but not necessarily the input data themselves) for the distance calculations, as supplementary information.

The merit of this paper is both conceptual and methodological.

Providing open access to code increases impact and citations, and allows analyses to be replicated.

The current option given (to request code from authors) is not typically viewed as sufficient, at least in open-access publishing.

Reviewer #3 (Remarks to the Author):

I am more than satisfied with the revised version. This is a significant contribution to the literature.

Response to Reviewers

Reviewer #1:

My comments here reflect only a divergence of opinion on the interpretation of the proposed metrics, rather than their implementation.

I contended that the trajectory following approach would be mostly if not wholly applicable to leading edges of ranges, and the counterargument does not entirely satisfy. Granted not all trailing edge populations will be adjacent to core range populations and that some travel would be needed for those population members to traverse unfavorable habitat. But fundamentally the processes of retreat differ from those of population advance, and unless we are talking about mobile organisms, this is the difference between local colonization (dispersal) and extinction (death, reproductive failure).

It is also an interesting idea to look at these metrics from the viewpoint of their utility as measures of exposure to climate change, rather than their ability to predict actual shifts in ranges. But that makes them much less useful and almost impossible to validate with any real data.

The focus of the metrics is on range edges where all organisms are on the brink of their existing tolerance irrespective of the width of their thermal niche. Traversing a range of climates that are outside the thermal niches is as likely to impact species with narrow ranges as those with broad ones. I respectfully disagree that thermal range width is relevant to the vulnerability to the effects of isolation (high MCE values). Communities comprised of narrow ranged species are vulnerable to climate change wherever they are, isolated or not.

We thank the reviewer for his or her comment. We would note that from the first sentence of the introduction and throughout the ms we unequivocally state that our approach is a coarse-filter approach aimed at developing climate change exposure metrics in the same manner that preceding climate velocity metrics have been presented (e.g. Loarie et al. 2009, Hamann et al. 2015). We clearly highlight the distinction between our approach and potential adaptations which focus on individual species in lines 186-207 in the current revision of the ms. We do not agree that the “focus of the metrics is on range edges where all organisms are on the brink of their existing tolerance irrespective of the width of their thermal niche.” This perspective is a species-specific approach and as noted by the reviewer applies to sessile organisms such as plants. Our approach is intentionally a coarse-filter approach which does not apply separately to mobile or sessile organisms but instead applies more broadly to biotic communities at a given site. The assumptions of this coarse-filter approach are relevant to previous efforts to produce climate velocity estimates and the implications of these assumptions (which we highlight on lines 189-194) have been elaborated on in earlier studies (e.g. Carroll et al. 2015). Moreover, we do not agree that developing an exposure metric is less useful or impossible to validate. All climate change exposure metrics are spatially explicit hypothesis. Previous studies have shown that in some cases these hypotheses are consistent with observed biotic patterns (e.g. Sandel et al. (2011) and Pinsky et al. (2013)). Further, our intention is not to use the trajectories we create to predict actual shifts in ranges. This also would more appropriately be considered through the use of species-specific biotic velocities. Under the more narrow set of conditions identified by the reviewer (i.e. forward biotic velocities for sessile organisms) we agree that colonization would not occur outside the thermal niche of a given species and we are not suggesting that plants could colonize areas that are unsuitable for that species. What we are suggesting is that a site comprised of species with narrow climate tolerances will effectively have a higher MCE value because more of the neighboring landscape will be ‘unsuitable’ for those organisms. This is demonstrated in supplementary table 1; mean MCE increases as climate bin width decreases. We highlight this interpretation by adding similar language to the discussion (lines 191 to 195). To conclude, because a species-specific approach is not the focus of our study, we contend that the reviewers’ criticisms are not entirely applicable to the interpretation of the exposure metrics we present here.

Reviewer #2:

The authors have adequately addressed comments in the initial reviews, resulting in a very useful contribution to the field.

My sole remaining suggestion is that the authors provide a sample R script (but not necessarily the input data themselves) for the distance calculations, as supplementary information.

The merit of this paper is both conceptual and methodological. Providing open access to code increases impact and citations, and allows analyses to be replicated.

The current option given (to request code from authors) is not typically viewed as sufficient, at least in open-access publishing.

Thank you for the positive feedback. We are making the R scripts publicly available along with the gridded outputs from the study.

Reviewer #3

I am more than satisfied with the revised version. This is a significant contribution to the literature.

Thank you for the positive feedback.